# Leucine-Rich Diet Modulates the Metabolomic and Proteomic Profile of Skeletal Muscle during Cancer Cachexia

**DOI:** 10.3390/cancers12071880

**Published:** 2020-07-13

**Authors:** Bread Cruz, André Oliveira, Lais Rosa Viana, Leisa Lopes-Aguiar, Rafael Canevarolo, Maiara Caroline Colombera, Rafael Rossi Valentim, Fernanda Garcia-Fóssa, Lizandra Maia de Sousa, Bianca Gazieri Castelucci, Sílvio Roberto Consonni, Daniel Martins-de-Souza, Marcelo Bispo de Jesus, Steven Thomas Russell, Maria Cristina Cintra Gomes-Mardondes

**Affiliations:** 1Department Structural and Functional Biology, Institute of Biology, UNICAMP, Campinas, 13083-862 São Paulo, Brazil; bread.cruz@gmail.com (B.C.); ago_oliveira@yahoo.com (A.O.); lala.viana311088@gmail.com (L.R.V.); leisaaguiar@yahoo.com.br (L.L.-A.); colomberamaiara@gmail.com (M.C.C.); rafaelrossiphd@gmail.com (R.R.V.); 2Department Cancer Physiology, H. Lee Moffitt Cancer Center and Research Institute, Tampa, FL 33612, USA; rafaelcanevarolo@gmail.com; 3Nano-Cell Interactions Lab, Department Biochemistry and Tissue Biology, Institute of Biology, UNICAMP, Campinas, 13083-862 São Paulo, Brazil; fefossa@gmail.com (F.G.-F.); dejesus@unicamp.br (M.B.d.J.); 4Department of Biochemistry and Tissue Biology, Institute of Biology, UNICAMP, Campinas, 13083-862 São Paulo, Brazil; l172421@dac.unicamp.br (L.M.d.S.); biancastelucci@gmail.com (B.G.C.); consonni@unicamp.br (S.R.C.); 5Laboratory of Neuroproteomics, Department of Biochemistry and Tissue Biology, Institute of Biology, University of Campinas, Campinas, 13083-862 São Paulo, Brazil; dmsouza@unicamp.br; 6Experimental Medicine Research Cluster (EMRC), University of Campinas, 13083-862 São Paulo, Brazil; 7D’Or Institute for Research and Education (IDOR), 04501-000 São Paulo, Brazil; 8School of Bioscience, College of Health and Life Sciences, Aston University, Birmingham B4 7ET, UK; s.t.russell1@aston.ac.uk

**Keywords:** leucine-rich diet, experimental cachexia, mitochondria dysfunction, metabolomic, proteomic, skeletal muscle

## Abstract

*Background:* Cancer-cachexia induces a variety of metabolic disorders, including skeletal muscle imbalance. Alternative therapy, as nutritional supplementation with leucine, shows a modulatory effect over tumour damage in vivo and in vitro. *Method:* Adult rats distributed into Control (C), Walker tumour-bearing (W), control fed a leucine-rich diet (L), and tumour-bearing fed a leucine-rich diet (WL) groups had the gastrocnemius muscle metabolomic and proteomic assays performed in parallel to in vitro assays. *Results:* W group presented an affected muscle metabolomic and proteomic profile mainly related to energy generation and carbohydrates catabolic processes, but leucine-supplemented group (WL) recovered the energy production. In vitro assay showed that cell proliferation, mitochondria number and oxygen consumption were higher under leucine effect than the tumour influence. Muscle proteomics results showed that the main affected cell component was mitochondria, leading to an impacted energy generation, including impairment in proteins of the tricarboxylic cycle and carbohydrates catabolic processes, which were modulated and improved by leucine treatment. *Conclusion*: In summary, we showed a beneficial effect of leucine upon mitochondria, providing information about the muscle glycolytic pathways used by this amino acid, where it can be associated with the preservation of morphometric parameters and consequent protection against the effects of cachexia.

## 1. Introduction

Cachexia is a multifactorial syndrome that occurs in patients with cancer, heart failure, rheumatoid arthritis, obstructive pulmonary disease, and during chronic infections [1,2]. Approximately 9 million patients globally are affected by cachexia, and it is considered an indicative risk factor for death [3,4]. This syndrome leads to severe host tissue wasting and intense body weight loss, mainly due to both muscle mass loss and adipose tissue decrement. In the late stage of cancer, cachexia is considered to be one of the most distressing phases, which causes not only functional impairment and psychological anguish but also decreases the anticancer treatment responses [5]. It is important to consider multiple factors while developing treatment options for cancer cachexia to maintain the quality of life of these patients [6]. Therefore, studies have been searching for better knowledge about the cachexia-induced molecular alterations on metabolic processes [7,8]. These studies about the drugs, nutritional supplementation and physical activity are working out to understand better how they could help and minimise the cachexia state, mainly focusing the lean body mass.

Accordingly, coadjuvant treatments such as nutritional supplementation with branched-chain amino acids could be an alternative to attenuate the effects of cachexia. More specifically, leucine has shown some beneficial impacts on reversing the severe damage caused by cachexia, mainly by the effective action in the maintenance of muscle protein synthesis and reducing proteolysis [7,8,9,10,11]. Indeed, there is an increased number of studies about nutritional supplementation in cachectic models [12,13], especially using leucine, where the exact mechanism and molecular action in cachexia remains unclear [14]. However, recent works show that the cachexia state affects the muscle energy generation throughout mitochondria, and leucine supplementation could modulate this process [15].

Mitochondria is the central energy-generating organelle in eukaryotic cells, and their imbalances cause profound changes to cells, especially to glycolytic pathway performance changing the whole cellular function [16,17]. Recent studies have shown that mitochondria are an essential framework for understanding the cachexia process [18], highlighting that during cancer-cachexia, there is both an imbalance in skeletal muscle mitochondrial biogenesis [18] and an increase in mitochondrial apoptosis [19]. Therefore, the use of an alternative nutritional scheme that supports the integrity and activity of mitochondria may be useful to maintain the lean mass in cachectic patients.

In the last few years, studies focused on physiology and damages of muscle mitochondria during cancer-induced cachexia [18,20,21]. Therefore, it is appropriate to investigate new evidence regarding the mechanism by which amino acids, particularly leucine, could improve the metabolic pathways protecting the mitochondrial function, especially in rat muscle by accessing an experimental model of cachexia such as Walker-256 tumour. Thus, it is known that the Walker-256 tumour grows in an exponential pattern, killing the bearing rat approximately 25 days after the tumour implantation; then, when there was a 3% leucine supplementation, a significant interference could be observed in cell signalling and modulation of mTOR in host tissues, especially in the gastrocnemius muscle [14]. In this way, to investigate many metabolic pathways, we used the metabolomic and proteomic analyses to bring information about the presence of specific metabolites and proteins in muscle tissue of tumour-bearing groups, subjected to nutritional supplementation with leucine. In the present study, we found that leucine supplementation modulated pathways that favoured mitochondrial biogenesis in skeletal muscle tissue and cell, maintaining the energy production activity. Therefore, the improvement of mitochondrial function during tumour development by using a leucine-rich diet contributes to mitochondrial enzymes and proteins related to the generation of more energy for the muscle tissue, minimising the effects of an experimental model of cachexia.

## 2. Results

### 2.1. Tumour-Induced Morphometric Damages Time-Course Dependent Were Modulated by Leucine-Rich Diet

Our results showed that both tumour-bearing groups (W and WL) had decreased body weight gain along experimental time in comparison to their respective control group (Table 1). However, the body weight gain in WL was significantly higher than in the W group (Table 1). The tumour evolution imposed a damaging effect on muscle weights at 14th and 21st days only for the W group. The tumour weight had no change in both groups, showing an exponential growth curve. The lean body mass reduced in both tumour-bearing groups, but was more pronounced in W group, at 14th and 21st days. Body fat percentage decreased in both tumour group at 21st day, associated with an increased body water percentage in both tumour-bearing groups, describing the characteristic process of cachexia state in this experimental model of cachexia. This process happened in a dependent time-course (Table 1).

### 2.2. Tumour Growth Induced Metabolomic Profile Changes in Skeletal Muscle

Among 42 skeletal muscle metabolites that were identified (Appendix A), five were altered in rats euthanatised at 7th and 21st days of the experiment (Figure 1). Comparing both tumour-bearing groups with control groups, at day 7, glycerol increased in W7 in comparison to C7 group (Figure 1a) and leucine and methionine enhanced in WL7 in comparison to C7 and W7 groups (Figure 1b,c). At the 21st day, muscle carnosine content decreased in WL21 in comparison to C21 group (Figure 1d), but glucose-6-phosphate increased in WL21 in comparison to L21 and W21 groups (Figure 1e) and WL21 group also had a higher methionine content in comparison to C21, L21 and W21 groups (Figure 1f).

During the tumour evolution (Appendix A), seven metabolites were modified in rats euthanatised at 7th, 14th and 21st days of the experiment as a time-course dependent related with/without modulation by leucine-rich diet (Figure 2). The muscle ADP content increased in W21 in comparison to W7 group but did not change in WL21 group (Figure 2a). Carnosine content in muscle was unchanged in the W group but deeply decreased in WL21 in comparison to earlier tumour growth in WL7 and WL14 groups (Figure 2b). Creatinine decreased in W21 when compared to the W14 group, and leucine-rich diet did not influence this metabolite (Figure 2c). In W group, the muscle glycerol content decreased at 14th and 21st days of tumour evolution, but in leucine-treated groups, this decrease was observed only in WL21 in comparison to WL7 group (Figure 2d). Leucine-rich diet influenced the muscle content, increasing the glycine and decreasing the leucine contents in WL14 and WL21 in comparison to WL7 group (Figure 2e,f). Additionally, muscle NAD+ content increased in W and WL groups at 14th day when compared to W7 and WL7 groups but decreased in WL21 in comparison to WL14 (Figure 2g).

### 2.3. Walker Factor-Induced Damages in C_2_C_12_ Myotubes Were Modulated by Leucine Treatment

In in vitro assays with C_2_C_12_ myotubes, we observed that both groups treated with WF and WFL presented lower cell proliferation than groups C and L (Figure 3a). However, the leucine-supplemented medium improved this process showing that the WFL group had an enhanced cell proliferation in comparison to W group (Figure 3a). Although the Walker Factor induced lower cell proliferation, the mitochondria activity increased in leucine-treated groups (Figure 3a), which is corroborated by the higher proportion of mitochondria in C_2_C_12_ myotubes treated with leucine (ultrastructure images, and graphic showed in Figure 3b–transmission electron microscopy, mitochondria numbers and area). Oxygen consumption showed that leucine-treated C_2_C_12_ myotube presented a higher respiratory rate (maximal mitochondrial consumption and spare respiratory capacity) even under the presence of the Walker Factor (Figure 3c). These results indicated that leucine treatment could improve the oxygen consumption associated with an increased ATP production in association with an increased number of mitochondria (Figure 3b).

Among 32 myotube metabolites that were identified (Appendix A), seven were altered in C_2_C_12_ cells (Figure 4). The leucine-treated group (WFL) had an increased aspartate in comparison to the WF group, as well as the choline content in contrast to C, L and WF groups (Figure 4a,b). Dimethylamine and glucose-6-phosphate were increased in WFL in comparison to C and WF groups (Figure 4c,d). Glutamine, N-Acetyllysine and sarcosine increased in WFL when compared to C, L and WF groups (Figure 4e–g).

### 2.4. Muscle Proteomic Profile Changed by Tumour Evolution but Was Diverted by Leucine-Rich Diet

Through proteomic analysis, we could compare the muscles of tumour-bearing rats without (W) and with leucine nutritional supplementation (WL), and realised that the presence of this amino acid in the diet altered mitochondrial proteins (34 different proteins in higher content in WL muscle than in W group, and three more mitochondrial proteins in lower content in WL about W group (Table 2). Additionally, in Table 3, the proteomic analysis could find changes in processes like molecular functions, biological processes and also the cellular components, when the comparison was made between both tumour-bearing groups, showing that the nutritional supplementation improved this functional component of the muscle cells. To display the different concentration of the identified proteins, we chose to compare the groups using the fold change values of each protein, always comparing one by one. We observed a significant increase in the concentration of enzymes that participate in processes for energy generation, such as malate dehydrogenase—Mdh1 (P93819) and Mdh2 (P04636) (Figure 5a,b, respectively; Table 2) among the leucine-treated groups. These two portions presented high concentrations influenced by leucine supplementation, in WL and L groups, when compared to the non-supplemented groups, showing that the nutritional supplementation could affect some point of the tricarboxylic acid cycle in these cachectic muscles. Glycogen-phosphorylase, Pygm (P09812) showed higher concentrations in both tumour-bearing groups, being more pronounced in the W group compared to the C, but less intense in the WL group compared to the L (Figure 5c; Table 2). Protein pyruvate dehydrogenase beta, Pdhb (P49432) appeared in higher concentrations in leucine-treated groups, besides the WL group had an increased content compared to the W group (Figure 5d; Table 2). Phosphoglucomutase-1, PGm1 (P38652; Table 2) had higher concentrations again in both leucine groups with or without tumour implant than the C and W groups (Figure 5e).

The present results showed throughout the proteomic analysis in muscle samples that the presence of the family of ATP synthases was in higher concentrations in the WL group compared to the W group (Table 2; Figure 6a). Since these proteins were deeply reduced in W group when compared to C group and slightly reduced in WL in comparison to L, we chose to present the graphic differences only between WL vs. W (see the differences in Appendix A). This proteins family belongs to the mitochondrial membrane and acts on the synthesis of ATP from ADP in the presence of a proton gradient across the membrane, which is generated by the electron transport complexes of the respiratory chain. Under the nutritional leucine supplementation, there was some modulation of this muscle ATP family, as presented in Figure 6a. Herewith, we observed changes in expression of flavoprotein (FP) subunit of succinate dehydrogenase (SDHa; Q920L2; Figure 6b) with higher values under leucine-rich diet, showing the modulatory effect, especially in tumour-bearing groups (Figure 6b). There was no significant difference between group W and C; C and L (see the results in Appendix A). Under leucine supplementation, the WL group presented the enzyme Creatine kinase M-type, Ckm (P00564) Ckm down-regulated in comparison to W (Figure 6b). On the other hand, creatine kinase S-type, Ckmt2 (P09605; Table 2, Figure 6b) had a higher concentration in WL compared to W group. Beta-enolase, Eno3 (P15429; P07323) was higher in WL group than the W group (Figure 6b).

Confirming the results obtained in proteomic analysis, the protein expression of some enzymes related to the mitochondria function was analysed by Western blotting. Cytochrome C oxidase (OXPHOS—Complex IV) shows a decreased expression in the W group when compared to the WL group, and against to the other control groups (Figure 6c), indicating changes in the accomplishment of oxidative phosphorylation and consequent reduction in energy production. Supporting this issue, the cyclic AMP responsive element-binding protein—Creb protein—involved with pathways between impaired mitochondria and the nucleus presented higher values in W compared to WL, indicating significant mitochondrial dysfunction (Figure 6c), being compatible with the higher muscle wasting presented in this group. Moreover, the WL group showed a lower tumour-induced muscle wasting which reinforced the benefits of leucine recovering the mitochondrial function.

## 3. Discussion

Cancer-induced cachexia is a complex systemic disease which is characterised by higher progressive weight loss and spoliation specifically of skeletal muscle which is reflected by intense and severe metabolic change crosstalking to other host tissues [22]. Cachexia is still a bad condition of these cancer patients, which depending on the stage, is rarely fully reversed [22,23]. Coadjuvant treatments are increasing to find a way to minimise the muscle atrophy related to cachexia [17]. In the present study, we found that leucine supplementation modulated pathways that favoured mitochondrial biogenesis in skeletal muscle tissue and cell, maintaining the energy production activity. In a cachectic state, this process showed a beneficial effect, minimising the deleterious effects of lean mass loss induced by tumour growth evolution, as presented here by the Walker tumour-induced muscle wasting. As seen here, the body weight gain, muscle weight and lean body mass were preserved in tumour-bearing animals which received leucine supplementation (WL) counteracting the highly wasting imposed by tumour-induced damages as observed in W group as a time-dependent way, starting on 14th day after tumour implant. This data corroborated with the literature, which shows that the tumour evolution is directly related to the spoliation of adipose tissue and decrease in muscle mass [13]. Another specific point related to cachexia state is oedema found in cancer patients and also in experimental animals [24]. Consistent with this fact, using the experimental model of cachexia—Walker-256 tumour—we observed deeply decrease in body fat and lean body mass was associated with higher body water content in both tumour-bearing independently of the nutritional scheme. As presented, we did not find any statistical differences in tumour weight evolution between both experimental groups. Despite having no increase in tumour mass, leucine supplementation stimulated the mTOR expression in Walker-256 tumours tissue, without additional growth [25]. As reported by Nakano and colleagues, leucine and other branched-chain amino acids increased cell senescence induced by DNA damage mediated by the mTORC1 pathway to regulate p21 translation [26], which could potentially explain the lack of increased proliferation in response to leucine supplementation. Moreover, considering the great importance in elucidating the effects of leucine supplementation in tumour growth, Viana and colleagues showed that the leucine diet led to a tumour metabolic shifting from glycolytic towards oxidative phosphorylation, reducing glucose consumption and metastasis formation, suggesting that the leucine nutritional supplementation had no benefit to the same experimental tumour model, as worked here [27]. Meanwhile, despite not increasing the tumour mass, leucine nutritional supplementation had also no benefit to this type of tumour, and as far as we know, clinical studies focused on leucine supplementation in cancerous tissues in patients are lacking, showing the limitations of an experimental model being translated to patients.

The morphometric alterations were accompanied by metabolic changes in the skeletal muscle tissue from both tumour-bearing groups. Both tumour-bearing groups exhibited alterations in some specific metabolites related to energy production, protein, and amino acid metabolism. Therefore, these metabolic alterations were mainly related to cachexia-induced muscle atrophy, as verified in vivo and in vitro assays [28]. These metabolic pathways may be involved in the high activity of skeletal muscle tissue [29]. Besides, the altered metabolites played a role in amino acid synthesis by reflecting the proteolysis occurred in skeletal muscle affected by the tumour growth effects, as tumour factors (Walker factor) and also in the whole host by the higher pro-inflammatory cytokines, as previously found [14].

For some of the omics experiments, there were limitations associated with a reduced sample number per group, therefore for some specific metabolites there was no significant differences. Higher levels of 3-methylhistidine, a product of peptide bond synthesis and the methylation of actin and myosin, were detected in both tumour-bearing groups later in the tumour progression. In this way, we chose to analyse the percentage of the content against the control group, and we verified that the W group started with higher muscle 3-methylhistidine at day 7 (4.4-fold; *p* = 0.0365) and this increased content remained on 14th (3.6-fold) and 21st days (2.9-fold), without showing significance (*p* = 0.2680 and *p* = 0.3380, respectively). On the other hand, the leucine-treated groups showed enhanced 3-methylhistidine content in muscle, independently of bearing a tumour or not (WL vs. L had 0.8-fold; 1.5-fold and 2.3-fold on 7th, 14th and 21st day, without showing statistical significance; *p* = 0.6763; *p* = 0.4713; and *p* = 0.9811, respectively). The increase in 3-methylhistidine provides an index for the rate of muscle protein breakdown [30,31,32]. Researchers have previously shown a positive correlation between increased 3-methylhistidine and cancer progression, along the cancer cachexia, due to the high muscle protein breakdown [32]. At present work, the W group started to increase 3-methylhistidine at 7th day, whereas in the leucine-treated rats had no difference. This point could be related to a prediction of cachexia starting and the leucine effects modulating the tumour-induced muscle protein breakdown and collaborating to the maintenance of host carcass.

Another point to discuss over the metabolomic results is that the reduction in muscle carnosine may be associated with a possible targeting of this metabolite to the tumour, since recent research pointed the potential physiological effects of carnosine exerts in all host tissues, and also in cancer cells [33,34]. Carnosine was related to the ability to inhibit glycolysis in tumour cells, by its carbonyl quenching ability and therefore reducing the generation of ATP [35,36]. In an exploratory view, we observed a slightly reduction of muscle carnosine in W21 group, and an expressive reduction in WL21 group. Despite this profound significance, showing that leucine potentiated the carnosine reduction, it could not be sustained as an anti-neoplastic action, because there is no difference in tumour mass in experimental groups. Most carnosine that reach the bloodstream is immediately hydrolysed into beta-alanine and histidine [36]. For this reason, carnosine was unable to reach the tumour and perform its antiproliferative role, thus further studies on muscle and tumour are needed to better understand the role of carnosine.

To mimic, in vitro, the effects of cachexia, observed in skeletal muscle tissue from Walker 256 tumour-bearing rats, we treated C_2_C_12_ myotubes with Walker Factor (WF) to evaluate the modulatory impact of leucine when muscle cells were exposed to WF. The WF, isolated from the ascitic fluid removed from a cachectic Walker 256 tumour-bearing rat [37], acts as the proteolysis-induced factor [38]. In the present study, we verified the proliferation of C_2_C_12_ cells exposed to WF and treated with leucine (WFL group) compared to the non-treated group (WF). This fact indicated that the presence of leucine stimulated mitoses [39,40], probably because there was a maintenance of the machinery responsible for energy production, keeping a physiological mitochondrial activity [17,41]. The maintenance of the mitochondrial activity and consequent oxidative phosphorylation (OXPHOS) metabolism were also observed in oxygen consumption rate, number and area of mitochondria in leucine-treated groups, even exposed the Walker Factor [42,43]. This fact indicated that mitochondrial function was likely preserved in WFL, showing the beneficial effect of leucine, which could be extended to the leucine-treated tumour-bearing animals. Corroborating the present results, our proteomic data showed that the muscle mitochondrial enzymes involved in the generation of energy through the tricarboxylic acid cycle (TCA) were in higher concentration as shown in the groups supplemented with leucine, especially in tumour-bearing group WL group when compared to the W group.

Moreover, our proteomic data showed that the W group had higher concentrations of proteins related to glycogen mobilisation and also stimulation of glycolysis. Therefore the higher Pygm found in W group likely was related to increased muscle glycogenolysis providing glucose for immediate use in glycolysis reactions. In contrast, the WL group could preserve the muscle glycogen (as previously showed [42,44] which could be in accordance with the maintenance of muscle energy source and consequently the muscle mass. Besides that, the Mdh1 and Mdh2 proteins that are related to mitochondrial catalytic activity and, together with the Pdhb protein, responsible for the conversion of pyruvate to Acetyl-CoA, showed high concentration in both groups supplemented with leucine, including the tumour-bearing group (L, WL). This fact indicated that the presence of this amino acid stimulated the production of energy through oxidative phosphorylation (OXPHOS) process.

In line with these results, our protein expression by Western blotting also revealed that the expression of mitochondrial proteins related to the OXPHOS complex was preserved in the leucine group. The oxidative phosphorylation electron transport chain complex is composed of five large protein complexes, Complex I—Complex V [43,45]. Indeed, the complexes I, III and IV are essential components of proton pump to maintain the electrochemical transmembrane potential which generates energy during oxidative phosphorylation; therefore, the expression of the complex IV that helped to produce ATP [46] was reduced in tumour-bearing animals (W) but remained unchanged in the WL group. Despite having no differences in muscle ATP content, as presented in metabolomic data, the W tumour-bearing group showed an increased muscle ADP content, which likely suggested a higher dissipation of energy, which did not happen in WL group. This result indicated that mitochondrial functional activity was preserved in the WL group. On the other hand, the W group showed a reduced concentration of mitochondrial proteins compared to the WL group. As verified by Arnould and colleagues, the mitochondrial activity impairment is sufficient to trigger CREB phosphorylation as seen in proliferation cells [47]. Although we were analysing muscle cells, the CREB activation suggested the fail in energy production in these cells. Besides, we found an increase in the phosphorylated CREB protein levels, indicating mitochondrial dysfunction or decreasing mitochondrial biogenesis in the W group. Moreover, once leucine supplementation altered the P-CREB activation in the WL group, we suggested that this amino acid could preserve the muscle energy production, indicating the protection of the mitochondrial machinery.

For this reason, the muscle tissue of tumour-bearing rats without leucine supplementation (W) needed to mobilise energetic molecules from other sources, such as muscle glycogen store and muscular glycolysis stimulation, as previously demonstrated by our group [48]. Additionally, the leucine group (WL) preserved these energy sources, and probably changed this glycolytic metabolism to better gain energy to muscle tissue supply.

ATP synthases complex is present in the mitochondrial inner membrane and acts on the synthesis of ATP from ADP in the presence of a proton gradient across the membrane, which is generated by the electron transport complexes of the respiratory chain. The proteins of this complex are essential in maintaining the correct mitochondrial energy production. The higher concentration of the found proteins (Table 2) found in the WL muscle tissue, as well the increase in maximal oxygen consumption and spare capacity in cultured myotubes, likely indicated that the respiratory chain activity was reestablished with the presence of leucine (Figure 7).

Other proteins involved in the mitochondrial catalytic activity, such as Pgm1, Sdha, Ckm/Ckmt2 and Eno3, also contribute to the understanding the mitochondrial activity, which was likely restored under leucine effects. Among them, the flavoprotein subunit of succinate dehydrogenase (Sdha), as the parting of the tricarboxylic acid cycle, was involved in complex II of the mitochondrial electron transport chain and is responsible for transferring electrons from succinate to ubiquinone (coenzyme Q). In line with our results, as seen by in vitro assay, leucine improved the maximal mitochondrial consumption and spare respiratory capacity, suggesting a more useful ability of energy generation in the WL muscle tissue than in W group. Besides, the higher concentration of Ckm protein in the W group suggested a current energy generation. However, as a counteracting effect of leucine, the higher Ckmt2 content found in the WL group likely indicated the restored mitochondrial activity to provide muscle energy. In parallel, Eno3 is an isoenzyme found in skeletal muscle cells, which plays a role in muscle development and regeneration [49] and also is related to catabolic glycolytic pathway. A switch from alpha enolase to beta enolase occurs in muscle tissue during development in rodents and also assures the mitochondria activity [50]. Corroborating with our results, the higher Eno3 concentration found in WL suggested the activation of energy production by preserving the mitochondria activity process more expressive than in W group.

Previously, we demonstrated that leucine supplementation increased the mTOR signalling, which was reflected in the maintenance of the total muscle protein net [51] (Figure 7). Moreover, a leucine-rich diet modulated pro- and anti-cachectic cytokines and the key proteins involved in the ubiquitin-proteasome pathway in gastrocnemius muscle tissue [14]. In tumour-bearing group fed a leucine-rich diet (the WL group) the maximum value of the cytokines was achieved at an earlier time point (on the 14th day) than in the W group. In parallel, this parameter was associated with an earlier increase in anti-inflammatory cytokines (IL-4 and IL-10), which could also have contributed to an earlier induction of the anti-inflammatory process, likely protecting the host against tumour-induced negative effects, thereby minimising the cachectic state.

This previously published work long with the new data presented here shows a relationship (Figure 7), because the increase in energy activity through mitochondrial proteins contributed to a better metabolic state and, consequently, stimulated processes of protection to the organism. Thus, leucine supplementation led to improvement and anticipated the anti-inflammatory responses, ameliorating the muscle protein synthesis and minimising the wasting. Since, we observed a modulatory effect over the ubiquitin-proteasome pathway [14], the increased expression in proteasome catalytic subunits likely indicated that the proteolytic process was activated in gastrocnemius muscle tissue in both tumour-bearing groups, but this activation was slightly less intense in the WL group. For this reason, we found in the present work that the action of ATP-synthase complex proteins is of a greater quantity in the leucine supplemented group, which improved the muscle energy production, together with the stimulation of protein synthesis signalling (increased mTOR signalling), resulting in an improvement of total muscle protein balance.

Summing up, these results contribute to better understand the beneficial effects of leucine for the cachectic host organism, especially to the skeletal muscle metabolism. Therefore, further studies are now ongoing in our laboratory to improve the understanding of the leucine effects in the skeletal muscle from cachectic hosts. The present work presented information that serves as a basis for new developments in this field of study. It is well established that in the cachexia state there is a significant decrease in the expression of genes involved in mitochondrial fusion, fission, ATP production and also mitochondrial density [15], leading to an impaired mitochondrial function.

## 4. Materials and Methods

### 4.1. In Vivo Assays

#### 4.1.1. Diets

The semi-purified isocaloric diets were in accordance to American Institute of Nutrition as a normal-protein diet (C), containing 18% protein, and leucine-rich diet (L), including 18% protein plus 3% L-leucine [52]. Both diets contained approximately 60% carbohydrate (sucrose, dextrin and starch), 7% fat (soybean oil) and 5% fibre (purified micro-cellulose). Vitamins and mineral mix, as well as cystine and choline, supplemented the diets. The control diet contained 1.6% L-leucine, and the leucine-rich diet contained 4.6% L-leucine, according to the protein source from casein, as used in our previous experimental studies [11]. Prior to these experiments dose-response assays were carried out with 1.5, 3.0, 4.5 and 6% leucine as a supplementation to the control diet, and all positive effects were found with 3% leucine supplementation shown in all morphometric and biochemical parameters including skeletal muscle [14,51]. Above 4.5% of leucine had no additional increase in positive effects.

Animals with cachexia, for both groups W and WL, had anorexia causing them to obtain a lesser diet than healthy animals. We conducted previous studies, with pair-fed groups, where we supplied the same amount of diet ingested by the groups with tumour to the other non-tumour-bearing groups, that is, the weight of the diet provided was normalised by the weight of the ingestion of the tumour group. Therefore, we have already validated in our laboratory that the amount of food ingested did not interfere with the parameters analysed in our experiments [14,51].

#### 4.1.2. Animal and Experimental Procedures

Male Wistar rats (*n* = 72 animals; 90-days-old, weighing 350–380 g) were obtained from the Centre/UNICAMP animal facilities (CEMIB/State University of Campinas, Brazil), and received food and water ad libitum under light-dark cycles (12/12 hours each) and constant temperature (22 ± 2 °C) and humidity (50–60%). The animals were distributed into 4 groups based on tumour implant (1 × 10^6^ viable cells of Walker-256 tumour injected in the right subcutaneous flank; cell viability obtained by trypan blue exclusion) and subjected or not to leucine-rich diet following the groups: 1. Control (C) subjected to a normoproteic diet; 2. Leucine (L) subjected to leucine-rich diet; 3. Tumour-bearing (W) subjected to a normoproteic diet, and implanted with Walker 256 carcinosarcoma cells; 4. Leucine tumour-bearing (WL) subjected to leucine-rich diet and implanted with Walker 256 carcinosarcoma cells. Each group contained a minimum of four animals for the metabolomic assay, and a minimum of six for the other analyses. The general UKCCCR [53] guidelines for animal welfare were followed, and the Institutional Committee for Ethics in Animal Research approved the study protocol (CEEA/IB/UNICAMP, protocol number #4918-1).

All animals remained in collective cages, and the body weight and tumour evolution were analysed every three days. After the 7th, 14th or 21st days of the experiment, all animals were randomised and euthanised on each day by decapitation. After euthanasia, to evaluate and establish trends regarding changes in metabolomics profiles in muscles and the effects of these changes in mitochondria, the gastrocnemius muscle and tumour was quickly dissected, weighed, frozen in liquid nitrogen and subsequently stored in −80 °C for further biochemical and molecular analyses. The metabolomics assays were accessed on 7th, 14th and 21st days. In our experimental conditions and also as an expensive assay, we chose to perform the proteomics assays only at day 21 as the results were more significant under the tumour effects and also leucine modulation.

### 4.2. In Vitro Assays

C_2_C_12_ myoblast cells from Cancer Research Laboratory (Aston University, Birmingham, UK) were grown in DMEM-high glucose medium (Sigma Chemical Company, St. Louis, MO, USA) supplemented with 10% foetal calf serum (FCS, Sigma) and 1% penicillin/streptomycin (P/S, Sigma) at 37 °C in a humidified 5% CO_2_ atmosphere. The myoblast cells, at 80% confluence, were differentiated into myotubes, using the propagation medium replaced by medium supplemented with 2% horse serum. Cells were assayed experimentally within 5 days of differentiation. Leucine-treated C_2_C_12_ myotubes were incubated with leucine (50 μM, Sigma) for 24 h [54]. Walker Factor-treated C_2_C_12_ myotubes were incubated with a protein named as Walker Factor, which was purified from ascitic fluid obtained from Walker-256 tumour-bearing rats (24 kDa), as described previously by Yano and colleagues [37], briefly, the ascitic fluid was centrifuged to remove the tumour cells, following filtration through ultra-centrifugal filter 10,000 MW cut off, and quantified by Western blotting using proteolysis-induction factor antibody (purified by Cancer Research Laboratory, Aston University, Birmingham, UK). C_2_C_12_ myotubes were incubated with Walker Factor for 24 h at final concentrations of 25 μg/mL [55]. Leucine/Walker Factor-treated C_2_C_12_ myotubes were first exposed to 50 μM leucine for 2 h, and then the cells were incubated with fresh DMEM medium leucine supplemented with Walker Factor at final concentrations of 25 μg/mL. These experiments followed the Control (C_2_C_12_C), Walker Factor (C_2_C_12_WF), Leucine supplementation medium (C_2_C_12_L) and, Walker Factor and leucine supplementation medium (C_2_C_12_WFL) groups.

### 4.3. Metabolomics Analysis

Muscle samples and C_2_C_12_ myotubes pellet were processed following Le Belle and colleagues’ protocol [56]. Briefly, about 100 mg of muscle tissue fragments or approximately 1 × 10^6^ cells were added to a cold methanol/chloroform solution (2:1 *v/v*, in a total of 2.5 mL) and sonicated (VCX 500, Vibra-Cell, Sonics & Material Inc., CT, USA) for 3 min with a 10-s pause interval between each min. A cold chloroform/distilled water solution (1:1 *v/v*, in a total of 2.5 mL) was then added to the samples. Samples were briefly vortexed (to form an emulsion) and centrifuged at 3.1 × 10^3^ g for 20 min at 4 °C. The upper phase (containing methanol, water and polar metabolites) was collected and dried in a vacuum concentrator (miVac Duo Concentrator, GeneVac, UK). The remaining solid phase was rehydrated in 0.6 mL of D2O-containing phosphate buffer (0.1 M, pH 7.4) and 0.5 mM of TMSP-d4. Samples were added to a 5-mm NMR tube for immediate acquisition, using a Varian Inova NMR spectrometer (Agilent Technologies Inc., Santa Clara, CA, USA) equipped with a triple resonance probe and operating at a 1H resonance frequency of 500 MHz and constant temperature of 298 K (25 °C). A total of 256 free induction decays were collected with 32-k data points over a spectral width of 16 ppm. A 1.5-s relaxation delay was incorporated between scans, during which a continual water presaturation radio frequency field was applied. Spectral phase and baseline corrections, as well as the identification and quantification of metabolites present in samples, were performed using Chenomx NMR Suite 7.6 software (Chenomx Inc., Edmonton, AB, Canada).

### 4.4. Proteomics Analysis

Proteomic analysis was performed in a two-dimensional nano UPLC (2D-RP/RP) in an Acquity UPLC M-Class System (Waters Corporation, MA, USA), which is connected in line to a Synapt G2-Si mass spectrometer (Waters Corporation). The mass spectrometer executed Data-Independent Acquisitions, employing more specifically the Ultra-definition Data-independent Mass Spectrometry (UDMS^E^) method [57]. Peptide loads were carried to separation in a nanoACQUITY UPLC HSS T3 Column (1.8 μm, 75 μm × 150 mm; Waters Corporation). Peptide elution was achieved using an acetonitrile gradient from 7% to 40% (*v/v*) for 95 min at a flow rate of 0.4 μL/min directly into a Synapt G2-Si. For every measurement, the mass spectrometer was operated in resolution mode with an m/z resolving power of about 40,000 FWHM, using ion mobility with a cross-section resolving power at least 40 Ω/ΔΩ. MS/MS analyses were performed by nano-electrospray ionisation in positive ion mode nanoESI (+) and a NanoLock Spray (Waters, UK) ionisation source. The lock mass channel was sampled every 30 sec. Proteins were identified, and quantitative data were processed by using dedicated algorithms and cross-matched with the UniProt rat proteome database, version 2015/11 (70,225 entries), with the default parameters for ion accounting and quantitation [58]. The databases used were reversed spontaneously during the database queries and appended to the original database to assess the false/positive identification rate. For proper spectra processing and database searching conditions, we used Progenesis QI for a proteomics software package with Apex3D, peptide 3D, and ion accounting informatics (Waters). This software starts with LC-MS data loading then performs alignment and peak detection, which creates a list of interested peptide ions (peptides) that are explored within Peptide Ion Stats by multivariate statistical methods; the final step is protein identity. The following parameters were considered in identifying peptides: 1. digestion by trypsin allowing one missed cleavage; 2. methionine oxidation was considered a variable modification and carbamidomethylation (C), fixed modification; 3. false discovery rate (FDR) less than 1%. These criteria rejected the not satisfied identifications. For the cellular component, biological processes, and molecular function of quantified proteins, we used the program DAVID 6.8 [59], Rat Protein Reference Database, and Reactome Pathways [60]. Proteins whose variation in abundance was statistically significant, STRING 10 [61] (considering high-confidence interactions) and KEGG program [62], were used for protein–protein interaction analyses and evaluations of functional enrichment in the network.

### 4.5. Oxygen Consumption Rate

The C_2_C_12_ myotube cells (1.25 × 10^6^ cells), cultured as described above and treated or not with Walker Factor and leucine-supplemented medium, for 24 h, were maintained in 6-well plates or 75-cm^2^ bottles after differentiation. The oxygen consumption rate (OCR) was monitored through Oroboros Oxygraph-2K adding sequentially 2 µM oligomycin A, 2 µM Carbonyl cyanide 3-chlorophenylhydrazone and 1 µM antimycin A. The DatLab software package (OROBOROS, Innsbruck, Austria) was used for data acquisition (2-sec-time intervals) and analysis. The OCR was expressed as pmol O_2_ × s^−1^ × cell.

### 4.6. Mitochondrial Activity and C_2_C_12_ Proliferation

To evaluate mitochondrial activity, C_2_C_12_ cells were plated in a 12-well plate until differentiation into myotubes. The myotubes were then treated as described by in vitro assay (Section 4.2), exposed to leucine (50 μM, Sigma) and with Walker Factor for 24 h. At the day of the treatment, one plate of untreated cells was previously read to determine the absorbance at time zero. After 24 h of treatment, MTT 3-(4,5-Dimethyl-2-thiazolyl)-2,5-diphenyl-2H-tetrazolium bromide (Sigma-Aldrich, MO, USA) solution was added to the cells at 0.5 mg/mL. After 1.5 h of incubation at 37 °C, 100 µL of DMSO was added to dissolve the formazan crystals. Finally, the absorbance was read using the microplate reader Cytation 5 (Biotek, Winooski, VT, USA) at λ = 570 nm.

To evaluate cell proliferation, the C_2_C_12_ cells were seeded in a 12-well plate and incubated for 24 h, as described above. Then, using the kit Cell proliferation Assay (Invitrogen, Waltham, MA USA), and following the manufactures instruction, the treated cells were incubated with EdU (5-Ethynyl-2′-deoxyuridine) (Invitrogen). After 24 h of treatment, the cells were fixed with 4% PFA (paraformaldehyde) for 15 min. The coverslips with permeabilised cells, with 0.5% Triton X-100, were reacted for EdU and stained. Nuclei were labeled with Hoechst 33342, and the slides were mounted with Mounting medium DAKO (Agilent, Santa Clara, California, CA, USA). The fluorescence microscope Cytation 5 (Biotek, Winooski, VT, USA) was used (excitation/emission wavelengths 337/447 nm for Hoechst 33342 and 647/794 nm for Alexa Fluor 647) and provided calculation of the percentage of cell proliferation.

### 4.7. Electronic Microscopy C_2_C_12_ Myotubes

A cell monolayer grown over a glass coverslip was fixed with 2.5% glutaraldehyde and 3 mM CaCl_2_ in 0.1 M sodium cacodylate buffer for 5 min at room temperature followed by 1 h of incubation on ice. After fixation, the samples were washed three times in 0.1 M sodium cacodylate and 3 mM CaCl2 solution and post-fixed with 1% osmium tetroxide in 0.1 M sodium cacodylate buffer and 0.8% potassium ferrocyanide for 1 h and en bloc stained in ice-cold 2% uranyl acetate overnight. The cells were dehydrated in ethanol on ice, ending with four changes of 100% ethanol at room temperature. The dehydrated cells were infiltrated in Epon resin. After four changes of resin solution, a fifth resin change was performed, and the dish was immediately placed in a lab oven at 60 °C to be polymerised for 72 h. Ultrathin sections were cut with a Leica Ultracut microtome, stained with 2% uranyl acetate and Reynold’s lead citrate, and then examined in a LEO 906-Zeiss transmission electron microscope (at the Electron Microscopy Laboratory of Institute of Biology, Campinas State University) using an accelerating voltage of 60 kV.

### 4.8. Western Blot

The gastrocnemius muscles were homogenised in buffer (Tris Base 100 mM, Na_4_P2O_7_ 10 mM, FNa 100 mM, Na_3_VO_4_ 1 mM, EDTA 10 mM, PMSF 2 mM, Aprotinin 0.1 mg/mL, Triton X-100 1%, pH 7.4), centrifuged, quantified as the total protein analysis [63]. Then, 40 µg muscle homogenate protein were resolved in SDS acrylamide gels (12%) and transferred to 0.22 µm nitrocellulose membrane and probed with antibodies against pCREB (Cell Signalling, 1:1000) and OXPHOS Complex I and IV (Abcam, 1:1000) and corrected by the loading controls vinculin (Cell Signalling, 1:1000), GAPDH (Cell Signalling, 1:1000) and α-tubulin (SigmaAldrich, 1:10,000). The membranes were then incubated with secondary antibodies (Cell Signalling, 1:10,000) conjugated with HRP. After chemiluminescence reaction using ECL reagent (Amersham), the protein bands densitometry analysis was performed using Image Capture (Amersham) followed the Gel-Pro II software (1.0 version, Silver Spring, Cybernetics, MD, USA).

### 4.9. Statistical Analysis

All results of morphometric parameters and metabolomic profile data in skeletal muscle and myotubes were expressed as the mean ± standard deviation (SD). All analyses were performed by one-way ANOVA followed by post-hoc Tukey as a multiple comparison test among all experimental groups and by Student’s *t-*test for comparison between two experimental groups (e.g., analysis of the tumour tissue in the W and WL groups). A value of *p* < 0.05 was considered significant [64]. The statistical analyses were performed using the software Graph Pad Prism 7.0 (Graph-Pad Software, Inc., San Diego, CA, USA).

The proteomics statistical evaluation of the data was accessed with unique peptides being normalised using Central Tendency and the absolute deviation of the adjusted median absolute deviation (MAD), which represents the Max Fold Change, showing the absolute numerical relationship of how many times the concentration of a protein is statistically higher when comparing one group with another. The peptides were extrapolated to their corresponding protein through RRollup using the Grubbs test, with a minimum of three peptides and *p* < 0.05. The statistical significance of dysregulated proteins, accessed by the z-score and the *p*-value associated with a 95% confidence level, were calculated using robust estimators such as the median and MAD.

## 5. Conclusions

The mitochondrial catalytic activity is increased in the groups that received a leucine-rich diet (L and WL). This catalytic activity helps to produce more energy for the muscle tissue throughout stimuli for pyruvate and acetyl-CoA synthesis. Therefore, the improvement of mitochondrial function during tumour development by the use of a leucine-rich diet contributes to mitochondrial enzymes and proteins related to the generation of more energy for the muscle tissue, minimising the effects of cachexia.

## Figures and Tables

**Figure 1 cancers-12-01880-f001:**
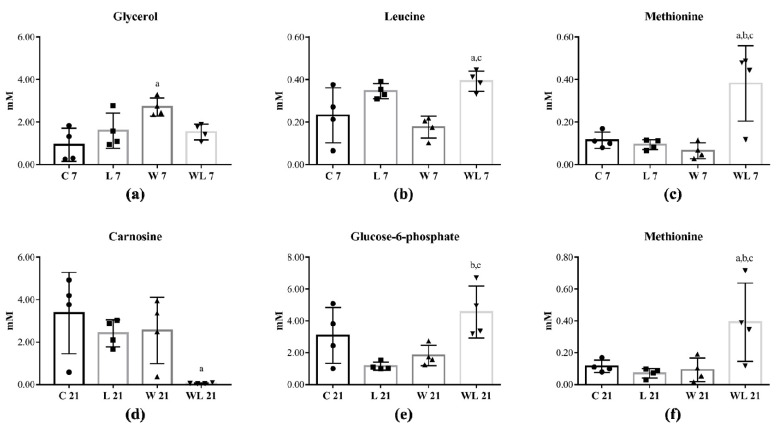
Skeletal muscle metabolic profile identified in rats euthanatised at 7th and 21st days of the experiment. Concentration of (**a**) Glycerol; (**b**) Leucine; (**c**) Methionine; (**d**) Carnosine; (**e**) Glucose-6-phosphate; (**f**) Methionine metabolites in different groups. Rats were distributed into control (C7 and C21); fed a leucine-rich diet (L7 and L21); Walker tumour-bearing (W7 and W21) and Walker tumour-bearing fed a leucine-rich diet (WL7 and WL21) euthanatised at 7th and 21st days of the experiment. Legend: C, control group; W, Walker tumour-bearing group; L, rats fed a leucine-rich diet; WL, tumour-bearing rats fed a leucine-rich diet. Data were expressed as mean ± standard deviation (SD) and analysed by one-way ANOVA followed by Tukey. (^a^) *p* < 0.05 comparison to C; (^b^) *p* < 0.05 comparison to L and (^c^) *p* < 0.05 comparison to W.

**Figure 2 cancers-12-01880-f002:**
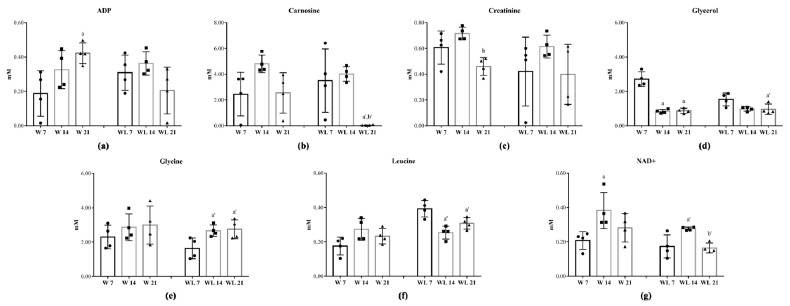
Skeletal muscle metabolic profile identified in rats euthanatised at 7th, 14th and 21st days of the experiment. Concentration of (**a**) ADP; (**b**) Carnosine; (**c**) Creatinine; (**d**) Glycerol; (**e**) Glycine; (**f**) Leucine; (**g**) NAD+. Rats were distributed into Walker tumour-bearing (W7, W14 and W21) and Walker tumour-bearing fed a leucine-rich diet (WL7, WL14 and WL21) euthanatised at 7th, 14th and 21st days of the experiment. Legend: C, control group; W, Walker tumour-bearing group; L, rats fed a leucine-rich diet; WL, tumour-bearing rats fed a leucine-rich diet. Data were expressed as mean ± standard deviation (SD) and analysed by one-way ANOVA followed by Tukey. (^a^) *p* < 0.05 comparison to W7; (^b^) *p* < 0.05 comparison to W14; (^a’^) *p* < 0.05 comparison to WL7 and (^b’^) *p* < 0.05 comparison to WL14.

**Figure 3 cancers-12-01880-f003:**
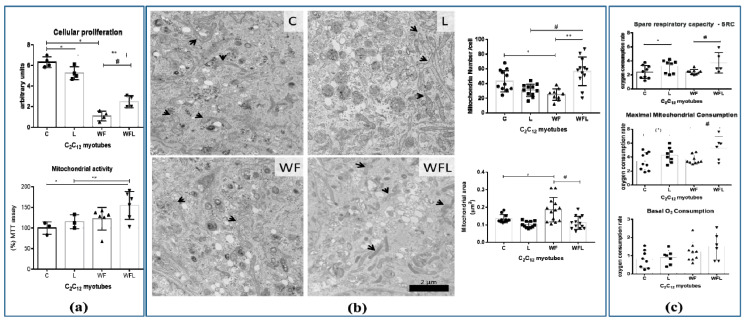
C_2_C_12_ myotubes parameters under Walker Factor effects and modulation by leucine treatment. (**a**) In vitro MTT assay for assessing cell mitochondrial activity in C_2_C_12_ myotubes subjected to Walker Factor and leucine supplementation. (**b**) Representative ultrastructure images of mitochondria from C_2_C_12_ myotubes. Groups without tumour presented a greater number of mitochondria, when compared to the other tumour-bearing groups (W and WL). The leucine supplemented tumour-bearing group (WL) had more mitochondria than the tumour-bearing group fed a control diet (W). Magnification 10000×. Scale bar: 2µm, arrows indicate mitochondria. Bar graphics show the mitochondrial number and area for each group. (**c**) Basal O_2_ mitochondrial consumption; maximal mitochondrial respiration and spare respiratory capacity of the C_2_C_12_ myotubes measured in Oroboros Oxygraph-2K and DatLab software package (OROBOROS, Innsbruck, Austria). Experimental design applied on myotubes cells were sequentially treated with Oligomycin A (1 µM), FCCP (0.5 µM), and Antimycin A (1 µM) as indicated by the arrows. For more details see Material and Methods section. Leucine-treated C_2_C_12_ myotubes were incubated with leucine (50 μM, Sigma) for 24 h. Tumour-treated C_2_C_12_ myotubes were incubated for 24 h with Walker Factor at final concentrations of 25 μg/mL. Leucine/Walker Factor-treated C_2_C_12_ myotubes were exposed to 50 μM leucine and supplemented with tumour Factor at final concentrations of 25 μg/mL. For more details see Material and Methods section. Legend: control (C); leucine supplementation medium (L); Walker Factor treatment (WF) and Walker Factor treatment and leucine supplementation medium (WFL). * *p* < 0.05 difference against C group; ** *p* < 0.05 difference against L group: # *p* < 0.05 difference against WF group.

**Figure 4 cancers-12-01880-f004:**
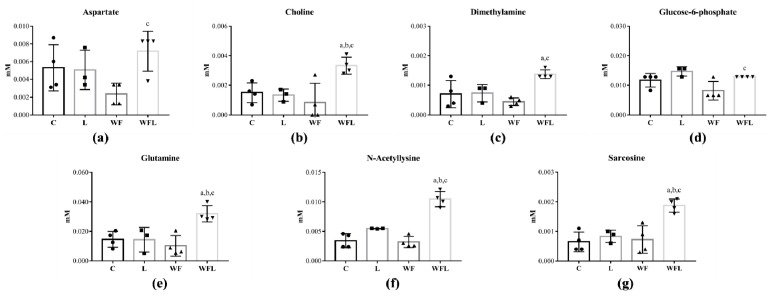
C_2_C_12_ Myotube metabolic profile identified in C_2_C_12_ cells. Concentration of (**a**) Aspartate; (**b**) Choline; (**c**) Dimethylamine; (**d**) Glucose-6-phosphate; (**e**) Glutamine; (**f**) N-Acetyllysine; (**g**) Sarcosine. C_2_C_12_ myotube cells were distributed into control (C); leucine supplementation medium (L); Walker Factor treatment (WF) and Walker Factor treatment and leucine supplementation medium (WFL). Data were expressed as mean ± standard deviation (SD) and analysed by one-way ANOVA followed by Tukey. (^a^) *p* < 0.05 comparison to C; (^b^) *p* < 0.05 comparison to L and (^c^) *p* < 0.05 comparison to W.

**Figure 5 cancers-12-01880-f005:**
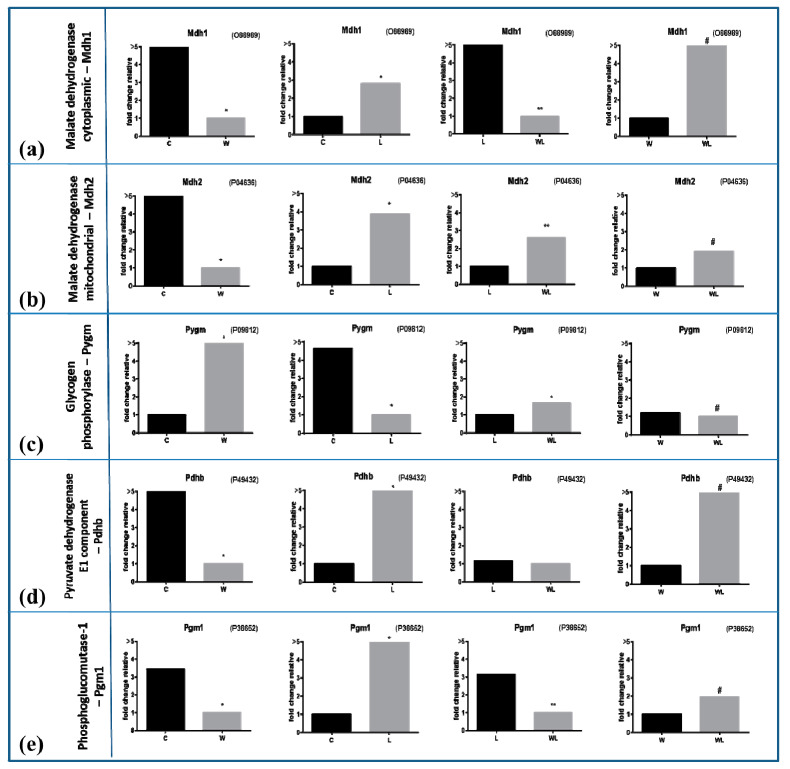
Muscle proteomic profile in all experimental groups subject or not to leucine-rich diet and bearing or not Walker-256 tumour. (**a**) malate dehydrogenase cytoplasmic—Mdh1, accession code O88989 and (**b**) malate dehydrogenase mitochondrial—Mdh2, accession code P04636 in gastrocnemius muscle under effects of Walker-tumour development and nutritional supplementation with leucine. (**c**) Glycogen phosphorylase—Pygm, accession code P09812 and (**d**) Pyruvate dehydrogenase E1 component—Pdhb, accession code P49432 in gastrocnemius muscle. (**e**) Phosphoglucomutase-1—Pgm1, accession code P38652 in gastrocnemius muscle. The values are expressed as the difference of maximal fold change. Legend: C, control group; W, Walker tumour-bearing group; L, rats fed a leucine-rich diet; WL, tumour-bearing rats fed a leucine-rich diet. * *p* < 0.05 difference against C group; ** *p* < 0.05 difference against L group: # *p* < 0.05 difference against WF group.

**Figure 6 cancers-12-01880-f006:**
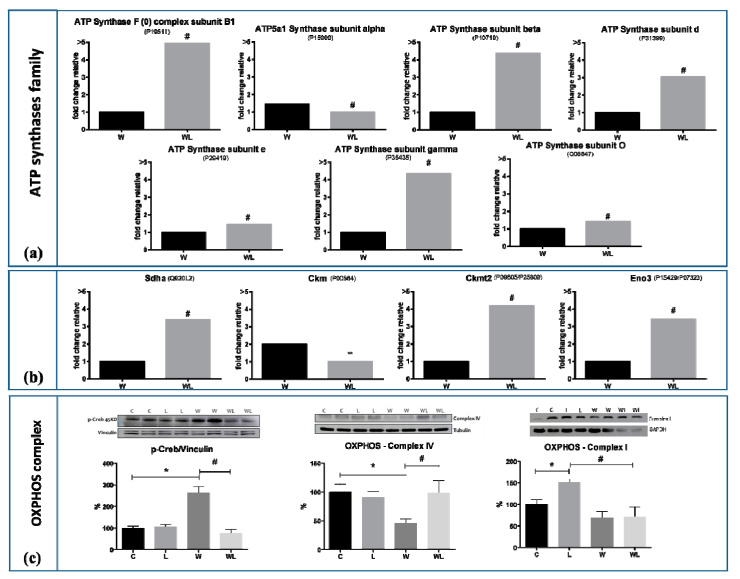
Muscle mitochondrial proteins of all experimental groups subject or not to leucine-rich diet and bearing or not Walker-256 tumour. (**a**) ATP Synthases family: average concentrations of the ATP synthase complex subunit B1 (accession code P19511), subunit alpha (accession code P15999), subunit beta (accession code P10719), subunit d (accession code P31399), subunit e (accession code P29419), subunit gamma (accession code P35435) and subunit O (accession code Q06647). (**b**) Succinate dehydrogenase [ubiquinone] flavoprotein subunit—Sdha (accession code Q920L2); Creatine Kinase M-type, cytoplasmic—Ckm (accession code P00564); Creatine Kinase S-Type, mitochondrial—Ckmt2 (accession code P09605/P25809); Beta-Enolase—Eno3 (accession code P15429/P07323). (**c**) OXPHOS proteins of Western blot assay: Protein CREB phosphorylated (p-Creb: 43 kDa; loading control vinculin 124 kDa), OXPHOS—Complex IV (40 kDa; loading control tubulin 55 kDa) and OXPHOS—Complex I (20 kDa; loading control GAPDH 37 kDA) in gastrocnemius muscle. Western blot images represent the representative from each group. Legend: C, control group; W, Walker tumour-bearing group; L, rats fed a leucine-rich diet; WL, tumour-bearing rats fed a leucine-rich diet. * *p* < 0.05 difference against C group; ** *p* < 0.05 difference against L group: # *p* < 0.05 difference against WF group. Detailed information about Western blot can be found at Appendix A.

**Figure 7 cancers-12-01880-f007:**
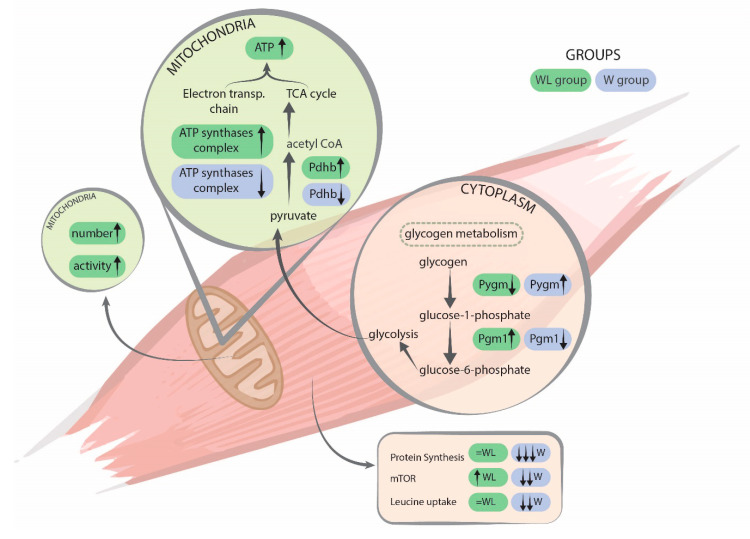
Illustration summarising the main processes found in both experimental groups. Legend: W, Walker tumour-bearing group, represented in blue; and WL, tumour-bearing rats fed a leucine-rich diet, represented in green colour. Arrows directions show the variation of the events between both groups. Glycogen phosphorylase (Pygm) is involved in muscle glycogenolysis and glycolysis. Phosphoglucomutase-1 (Pgm1) has a catalytic activity breakdown and synthesis of glucose. Pyruvate dehydrogenase E1 component (Pdhb) plays a role in the process of converting pyruvate to Acetyl-CoA. ATP synthases complex is for the production of ATP by the respiratory chain, mitochondrial function. Muscle protein synthesis and signalling events were published previously by Cruz and colleagues [51].

**Table 1 cancers-12-01880-t001:** Morphometric parameters of all experimental groups subjected or not to leucine-rich diet and tumour-bearing or control groups as a time-course analyses.

Morphometric Parameters	Group	7th Day	14th Day	21st Day
		Mean ± SD	Mean ± SD	Mean ± SD
**Body Weight Gain (g)**	C	7.72 ± 0.13	14.45 ± 0.18	27.78 ± 2.78
	L	7.84 ± 1.31	21.25 ± 2.78	34.10 ± 0.98
	W	7.00 ± 0.72	10.11 ± 1.96 *	14.06 ± 1.92 *
	WL	8.56 ± 0.67	13.67 ± 1.49 **^#^	22.13 ± 2.79 **^#^
**Muscle Weight (g)**	C	2.20 ± 0.20	2.20 ± 0.10	2.20 ± 0.20
	L	2.38 ± 0.33	2.31 ± 0.19	2.11 ± 0.13
	W	2.15 ± 0.19	1.78 ± 0.23 *	1.70 ± 0.16 *
	WL	2.48 ± 0.26 ^#^	2.04 ± 0.19 **^#^	2.02 ± 0.13 ^#^
**Lean Mass (%)**	C	n/e	23.39 ± 1.65	24.00 ± 1.41
	L	n/e	20.49 ± 5.00	23.00 ± 2.64
	W	n/e	11.92 ± 2.93 *	15.33 ± 3.56 *
	WL	n/e	15.28 ± 7.18 **	19.25 ± 3.78 **
**Body Fat (%)**	C	n/e	28.00 ± 1.56	29.70 ± 1.70
	L	n/e	36.63 ± 1.50	35.83 ± 2.45
	W	n/e	25.15 ± 3.40 *	23.75 ± 6.94 *
	WL	n/e	28.20 ± 3.12 **	21.20 ± 6.27 **
**Body Water (%)**	C	n/e	45.30 ± 2.20	43.00 ± 2.00
	L	n/e	39.43 ± 2.86	37.67 ± 0.33
	W	n/e	59.38 ± 1.58 *	56.50 ± 3.64 *
	WL	n/e	53.00 ± 2.88 **	56.00 ± 3.48 **
**Tumour Weight (g)**	C	-	-	-
	L	-	-	-
	W	11.76 ± 1.89	22.14 ± 4.24	35.04 ± 3.49
	WL	15.22 ± 2.05	24.76 ± 2.89	38.8 ± 2.23

Body weight gain corresponds to the difference of final body weight minus the previous body weight per day (g). Absolute gastrocnemius muscle weight (g). Percentage of lean mass and body fat measured by DEXA analyses. Percentage of total body water obtained by constant weight of dry carcass. Absolute tumour weight (g). For details see Material and Method section. Legend: C, control group; W, Walker tumour-bearing group; L, rats fed a leucine-rich diet; WL, tumour-bearing rats fed a leucine-rich diet; n/e, not evaluated. Values are means ± standard deviation (SD). * *p* < 0.05 difference against C group; ** *p* < 0.05 difference against L group; ^#^
*p* < 0.05 difference against W group.

**Table 2 cancers-12-01880-t002:** The highest and lowest mitochondrial proteins concentration identified in rats with Walker tumour-bearing fed/not fed a leucine-rich diet and euthanatised at 21st day of the experiment.

Accession	Description	Max Fold Change	Highest Mean Condition	Lowest Mean Condition
Q5XI78	2-oxoglutarate dehydrogenase, mitochondrial GN = Ogdh	1.805	WL	W
17764	Acetyl-CoA acetyltransferase, mitochondrial GN = Acat1	>5	WL	W
Q9ER34	Aconitate hydratase, mitochondria GN = Aco2	1.811	WL	W
P00507	Aspartate aminotransferase, mitochondrial GN = Got2	1.196	WL	W
P19511	ATP synthase F(0) complex subunit B1, mitochondrial GN = Atp5f1	>5	WL	W
P10719	ATP synthase subunit beta, mitochondrial GN = Atp5b	4.378	WL	W
P31399	ATP synthase subunit d, mitochondrial GN = Atp5h	3.048	WL	W
P29419	ATP synthase subunit e, mitochondrial GN = Atp5i	1.477	WL	W
P35435	ATP synthase subunit gamma, mitochondrial GN = Atp5c1	4.358	WL	W
Q06647	ATP synthase subunit O, mitochondrial GN = Atp5o	1.420	WL	W
Q8VHF5	Citrate synthase, mitochondrial GN = Cs	3.133	WL	W
P09605; P25809	Creatine kinase S-type, mitochondrial GN = Ckmt2	4.191	WL	W
P32551	Cytochrome b-c1 complex subunit 2, mitochondrial GN = Uqcrc2	1.853	WL	W
P10888	Cytochrome c oxidase subunit 4 isoform 1, mitochondrial GN = Cox4i1	3.775	WL	W
P11240	Cytochrome c oxidase subunit 5A, mitochondrial GN = Cox5a	2.577	WL	W
P12075	Cytochrome c oxidase subunit 5B, mitochondrial GN = Cox5b	>5	WL	W
P13803	Electron transfer flavoprotein subunit alpha, mitochondrial GN = Etfa	1.343	WL	W
P14408	Fumarate hydratase, mitochondrial GN = Fh	>5	WL	W
P10860	Glutamate dehydrogenase 1, mitochondrial GN = Glud1	2.046	WL	W
P56574	Isocitrate dehydrogenase [NADP], mitochondrial GN = Idh2	2.443	WL	W
P15650	Long-chain specific acyl-CoA dehydrogenase, mitochondrial GN = Acadl	1.182	WL	W
P04636	Malate dehydrogenase, mitochondrial GN = Mdh2	1.922	WL	W
P08503	Medium-chain specific acyl-CoA dehydrogenase, mitochondrial GN = Acadm	>5	WL	W
Q66HF1	NADH-ubiquinone oxidoreductase 75 kDa subunit, mitochondrial GN = Ndufs1	>5	WL	W
Q9R063	Peroxiredoxin-5, mitochondrial GN = Prdx5	>5	WL	W
P49432	Pyruvate dehydrogenase E1 component subunit beta, mitochondrial GN = Pdhb	>5	WL	W
P15651	Short-chain specific acyl-CoA dehydrogenase, mitochondrial GN = Acads	1.174	WL	W
P48721	Stress-70 protein, mitochondrial GN = Hspa9	>5	WL	W
Q920L2	Succinate dehydrogenase [ubiquinone] flavoprotein subunit, mitochondrial GN = Sdha	3.386	WL	W
B2GV06	Succinyl-CoA:3-ketoacid coenzyme A transferase 1, mitochondrial GN = Oxct1	1.576	WL	W
Q9Z0V6	Thioredoxin-dependent peroxide reductase, mitochondrial GN = Prdx3	1.458	WL	W
Q64428	Trifunctional enzyme subunit alpha, mitochondrial GN = Hadha	1.682	WL	W
Q60587	Trifunctional enzyme subunit beta, mitochondrial GN = Hadhb	2.138	WL	W
P45953	Very long-chain specific acyl-CoA dehydrogenase, mitochondrial GN = Acadvl	3.580	WL	W
Q99NA5	Isocitrate dehydrogenase [NAD] subunit alpha, mitochondrial GN = Idh3a	2.031	W	WL
P15999	ATP synthase subunit alpha, mitochondrial GN = Atp5a1	1.473	W	WL
P13437	3-ketoacyl-CoA thiolase, mitochondrial GN = Acaa2	2.441	W	WL

Rats were distributed into Walker tumour-bearing (W) and Walker tumour-bearing fed a leucine-rich diet (WL) euthanatised at 21st day of the experiment. Statistical analysis described in Methods, showing the significant value for *p* < 0.05.

**Table 3 cancers-12-01880-t003:** The most impacted molecular function, biological processes and cellular components in muscle proteomic analyses identified in rats with Walker tumour-bearing fed/not fed a leucine-rich diet and euthanatised at 21st day of the experiment.

**# Pathway ID**	**Molecular Function Pathway Description**	**Gene Count**	**FDR**
GO.0003824	Catalytic activity	35	1.79 × 10^−19^
GO.0016491	Oxidoreductase activity	17	2.23 × 10^−14^
GO.0015078	Hydrogen ion transmembrane transporter activity	10	1.50 × 10^−13^
GO.0043167	Ion binding	29	1.33 × 10^−12^
GO.0048037	Cofactor binding	11	1.09 × 10^−10^
GO.0009055	Electron carrier activity	7	1.71 × 10^−09^
GO.0050662	Coenzyme binding	9	6.65 × 10^−09^
GO.0003674	Molecular function	33	1.48 × 10^−08^
GO.0046872	Metal ion binding	20	1.48 × 10^−08^
GO.0005215	Transporter activity	14	2.68 × 10^−08^
**# Pathway ID**	**Biological Process Pathway Description**	**Gene Count**	**FDR**
GO.0006091	Generation of precursor metabolites and energy	20	6.99 × 10^−29^
GO.0044710	Single-organism metabolic process	34	3.18 × 10^−23^
GO.0055114	Oxidation-reduction process	24	3.18 × 10^−23^
GO.0019752	Carboxylic acid metabolic process	21	1.59 × 10^−19^
GO.0072350	Tricarboxylic acid metabolic process	10	1.54 × 10^−17^
GO.0044281	Small molecule metabolic process	23	2.07 × 10^−17^
GO.0006099	Tricarboxylic acid cycle	9	3.87 × 10^−16^
GO.0045333	Cellular respiration	11	8.45 × 10^−16^
GO.0046496	Nicotinamide nucleotide metabolic process	11	8.45 × 10^−16^
GO.0015980	Energy derivation by oxidation of organic compounds	12	1.31 × 10^−15^
GO.0006732	Coenzyme metabolic process	13	2.82 × 10^−15^
**# Pathway ID**	**Cellular Component Pathway Description**	**Gene Count**	**FDR**
GO.0005739	Mitochondrion	32	3.66 × 10^−31^
GO.0031966	Mitochondrial membrane	23	1.17 × 10^−26^
GO.0005740	Mitochondrial envelope	23	3.64 × 10^−26^
GO.0044444	Cytoplasmic part	42	1.17 × 10^−25^
GO.0005743	Mitochondrial inner membrane	19	1.29 × 10^−23^
GO.0044429	Mitochondrial part	22	1.00 × 10^−22^
GO.0043209	Myelin sheath	14	2.79 × 10^−19^
GO.0005737	Cytoplasm	39	1.46 × 10^−16^
GO.0043234	Protein complex	26	5.62 × 10^−15^
GO.0070062	Extracellular exosome	20	6.93 × 10^−15^
GO.0031090	Organelle membrane	24	1.04 × 10^−14^
GO.0031988	Membrane-bounded vesicle	22	9.29 × 10^−14^

False discovery rate (FDR). Statistical analysis described in Methods, showing the significant value for *p* < 0.05.

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
