# Peer review of "Leucine-Rich Diet Modulates the Metabolomic and Proteomic Profile of Skeletal Muscle during Cancer Cachexia"

_cancers, 2020, doi:10.3390/cancers12071880_

Round 1

Reviewer 1 Report

The manuscript entitled "Leucine-rich diet modulates the metabolomic and proteomic profile of skeletal muscle during cancer cachexia" by Cruz et al. is the topic of this review. Overall, the article presentation is outstanding and the experimental approach fully supports the conclusions provided. There are some issues with the variability of the experimental findings impacting some of the conclusions that could be made, however, sufficient data was provided that makes this work of significant value to the scientific community. This work has merit and is recommended for publication with minor revisions. These include:

Figure 5 - please add standard deviation bars to the figure

Figure 6 - please add standard deviation bars to the panels (a) and (b)

Of final note, an illustration in the Discussion section that summarizes the findings and/or model may be extremely helpful for the less-initiated readers. This should be strongly considered in a revised draft.

Author Response

Referee #1

We would like to thank Reviewer #1 for the time spent in the revision of our manuscript and for the valuable comments and suggestions. We are completely certain that the revised manuscript improved substantially after your comments and suggestions.  The changes made in the revised paper are highlighted in yellow.

The manuscript entitled "Leucine-rich diet modulates the metabolomic and proteomic profile of skeletal muscle during cancer cachexia" by Cruz et al. is the topic of this review. Overall, the article presentation is outstanding and the experimental approach fully supports the conclusions provided. There are some issues with the variability of the experimental findings impacting some of the conclusions that could be made, however, sufficient data was provided that makes this work of significant value to the scientific community. This work has merit and is recommended for publication with minor revisions. These include:

Referee comment: Figure 5 - please add standard deviation bars to the figure

Figure 6 - please add standard deviation bars to the panels (a) and (b)

Answer: The authors would like to thank the comments and we understand this point. Although, with respect to the standard deviation bars in these graphics, the values used to build the Figures 5 and 6 (a and b) come from proteomic analysis obtained from the Max Fold Change, presented in Figures as the “fold change relative”, which shows an absolute numerical relationship that represents how many times the concentration of a protein is statistically higher in a group compared to another and, for this reason, there is no standard deviation bars. We have included this information in the main text, highlighted in yellow in statistical section.

Referee comment: Of final note, an illustration in the Discussion section that summarizes the findings and/or model may be extremely helpful for the less-initiated readers. This should be strongly considered in a revised draft.

Answer: We agree with the referee on this suggestion, which can improve our work, showing an illustration that pointed the main finding in the whole work. We included the Figure 7, pointing the main enzymes/protein and metabolites changed by tumour evolution and modulated by leucine-rich diet.

Figure 7: Illustration summarizing some processes present in the discussion section in W and WL groups. Glycogen phosphorylase (Pygm) is involved in the muscle glycogenolysis and glycolysis. Phosphoglucomutase-1 (Pgm1) has the catalytic activity breakdown and synthesis of glucose. Pyruvate dehydrogenase E1 component (Pdhb) plays a role in the process for converting pyruvate to Acetyl-CoA. ATP synthases complex is for production of ATP by the respiratory chain, mitochondrial function.

Reviewer 2 Report

The authors demonstrated that in Walker tumor bearing rats, supplementation of 3% leucine improved muscle wasting and that energy generation process in the mitochondria improved by the diet modification through proteomic and metabolomic analysis. They also analyzed acute change in morphology and respiration as well as some metabolites in cultured muscle cells in the presence/absence of tumor derived factor and/or leucine.

If one leucine tablet a day is the answer for cancer cachexia, I would be very happy for such patients. But I saw many terminal cancer patients suffer from cachexia despite ample nutritional supplementation. Unfortunately, this manuscript failed to convince me that leucine is the answer for cancer cachexia. After publishing many articles on this subject, the authors must show some mechanistic insight into how leucine changed the muscle metabolism. The study, although putting significant efforts, is largely descriptive.

 Despite many previous publications using the similar diet by the author’s group, I am not convinced that extra-leucine prevented cachexia. If they supplemented with 3% glycine or 3% more non-essential amino acids what would happen? In addition, it is very important to know whether there is any dose-response relation between leucine dose and muscle preservation. How about the amount of chow consumed by each group? I know this might be too much but to generalize their results, I would like to see different tumor models other than Walker tumor.

 I do not see in vitro muscle cell experiments have relevance to their in vivo experiments. The former were to evaluate acute effects while the latter were analyses of chronic effects.

 In Introduction and/or Discussion they need to discuss their previous results that leucine supplementation modulated cytokines, mTOR signaling, and proteasome function, in the context of new data. 

 The most striking difference this diet regimen made is that carnosine is disappeared from muscle in 21 days only in WL group. I do not think this is good for muscle cells. Please explain.

The authors did not describe proliferation assay and MTT assay (Fig 3a) in Method.

Author Response

Referee #2

Comments and Suggestions for Authors

The authors demonstrated that in Walker tumor bearing rats, supplementation of 3% leucine improved muscle wasting and that energy generation process in the mitochondria improved by the diet modification through proteomic and metabolomic analysis. They also analyzed acute change in morphology and respiration as well as some metabolites in cultured muscle cells in the presence/absence of tumor derived factor and/or leucine.

We would like to thank Reviewer #2 for the time spent in the revision of our manuscript and for the valuable comments and suggestions. We are convinced that the revised manuscript improved substantially after your comments.  The changes made in the revised paper are highlighted in yellow

Referee comment: If one leucine tablet a day is the answer for cancer cachexia, I would be very happy for such patients. But I saw many terminal cancer patients suffer from cachexia despite ample nutritional supplementation.

Answer: We do agree with the referee that would be wonderful to have one single tablet or supplementation which could reverse the cachexia state. Although we are trying to show some alternative view, and focusing this point to an experimental model, which in future could show some potential for patient treatment.  We would be very happy if we could find and show an effective treatment for cancer cachexia, however, we know that it is a complex picture and it must be approached in a multidisciplinary way. Our goal is to contribute with some explanations and description of how the cachexia mechanism works and if there are ways to modulate these mechanisms. For this reason, we use an amino acid that is already established as a protein synthesis signalling and we work with time-course experiments to describe some questions about when nutritional supplementation can be a supporter to the multidisciplinary approach in patient care. We are not proposing to treat with leucine the terminal patient, but we do propose the use of leucine in pre-cachectic stages in support of available therapies.

Referee comment: Unfortunately, this manuscript failed to convince me that leucine is the answer for cancer cachexia.

Answer: As explained in the previous answer, it is not our goal to use leucine to completely reverse the cancer cachexia, so the article is not intended to elect a treatment against cachexia, but to explain how this amino acid acts in skeletal muscle of a host with cancer.

Referee comment: After publishing many articles on this subject, the authors must show some mechanistic insight into how leucine changed the muscle metabolism. The study, although putting significant efforts, is largely descriptive.

Answer: We believe that it is important for this research area to use molecular tools to broaden the understanding of something as complex as cancer cachexia. Our aim is to describe some metabolites and metabolic pathways that are affected in this condition. This serves as a basis for future research and assists in the design of new assays. The objective is to build knowledge from descriptive observations, as a base for in future show mechanisms of action. However, our group has already presented some mechanisms in other works, such as Cruz, et al. 2017; Viana et al. 2016; Cruz et al. 2019; Viana et al, 2019. The present work described the effect of leucine in a time-course evolution of the cachexia, suggesting a disruptive approach when it shows possible metabolites and / or impacted metabolic pathways, opening the field for new works.

Referee comment: Despite many previous publications using the similar diet by the author’s group, I am not convinced that extra-leucine prevented cachexia.

Answer: Based on our many previous publications, we can state that leucine modulates protein synthesis and protects lean mass in cachexia, in our experimental conditions, as well as other works show this (Choudry et al. 2006; Garlick 2005; Gonçalves, E.M. Gomes-Marcondes 2010; Kimball and Jefferson 2001). However, preventing cachexia entirely is a utopia, we know that the sum of efforts, such as the use of medication, supplementation, physical exercises, among others, is capable of minimising cachectic effects, but even so, it does not totally prevent cachexia.

Referee comment: If they supplemented with 3% glycine or 3% more non-essential amino acids what would happen? In addition, it is very important to know whether there is any dose-response relation between leucine dose and muscle preservation.

Answer: There are studies that have shown that among the BCAAs - leucine, isoleucine and valine - leucine presents the effective modulation on protein synthesis, while the others do not show this activity (Garlick 2005). Based on our work, we chose leucine as an amino acid with important characteristics that can interfere in the cachexia process. Other amino acids could have been used, but due to the vast literature on the action of leucine, over the protein synthesis, interleukins, inflammatory processes, etc., this amino acid leucine is a strong candidate for cancer cachexia study. Additionally, our team have made essays with dose-response and the findings were that above 3% there is no additional benefit over muscle protection.

Referee comment: How about the amount of chow consumed by each group?

Answer: In this work we did not mention this parameter, but the animals with cachexia, both groups W and WL, had anorexia getting less diet than normal animals. We conducted previous studies, with pair fed groups, where we supplied the same amount of diet ingested by the groups with tumour to the other non-tumour-bearing groups, that is, the weight of the diet provided was normalised by the weight of the ingestion of the tumour group. Therefore, we have already validated in our laboratory that the amount of food ingested does not interfere with the parameters analysed in our experiments (Cruz, et al, 2017; Cruz, et al. 2019).

Referee comment: I would like to see different tumor models other than Walker tumor.

Answer: We have made some experimental essay with MAC16 subjected to leucine-rich diet analysing other parameters, instead of metabolomic and proteomic profile, and also in the literature there are other studies related with leucine supplementation (Peters et al. 2011; Viana and Gomes-Marcondes 2015), showing beneficial action of leucine.

Referee comment: I do not see in vitro muscle cell experiments have relevance to their in vivo experiments. The former were to evaluate acute effects while the latter were analyses of chronic effects.

Answer: Our approach was to work in vivo and in vitro to obtain as much information as possible from the association of leucine and cachexia. In this line, we mimic the cachexia situation using the Walker factor analysing the metabolites, which are most impacted in an environment without organism interference, as in muscle cell culture. In parallel, we performed in vivo tests to trace a proteomic profile during leucine supplementation in cachectic animals and this could allow us to make comparisons between isolated cells and complete organisms, even using an acute model comparing with the chronic effects. According to the proposal, the objective is to provide new information on impacted pathways and metabolites, studies like this allow new approaches to be envisioned, as happened recently with the growing interest in mitochondrial activity during cachexia or the mobilization of adipose tissue early by the cachectic organism (Hardee, Montalvo, and Carson 2017; Kliewer et al. 2015; Vaitkus and Celi 2017).

Referee comment: In Introduction and/or Discussion they need to discuss their previous results that leucine supplementation modulated cytokines, mTOR signaling, and proteasome function, in the context of new data. 

Answer: We fully agree with this comment, so we inserted a segment about this comparison in the discussion section of the article. This point is highlighted in yellow in the revised manuscript.

Referee comment: The most striking difference this diet regimen made is that carnosine is disappeared from muscle in 21 days only in WL group. I do not think this is good for muscle cells. Please explain.

Answer: We would like to thank the referee for raising this point. Although the value of carnosine is very low in WL group, almost “disappearing” (mean ± SD: 0.0514 ± 0.0248) in comparison to C21 (mean ± SD: 3.3680 ± 1.9179) group showing a significant reduction (Figure 1d and Table3), it is important and we are going to related this point including a brief discussion.  

The carnosine is a multifunctional histidine-containing dipeptide - formed by beta-alanine and histidine - that is abundant in high concentrations in mammalian skeletal muscle (Matthews et al., 2019; Perim et al., 2019). The potential roles of carnosine in skeletal muscles cells are: 1) proton buffering capacity; 2) regulator of calcium release and calcium sensitivity; 3) protection against reactive oxygen species; 4) chelation of transition metal ions and 5) extracellular provider of histidine/histamine (Boldyrev et al., 2013). In addition, recent research points towards a potential for carnosine to exert a wider range of physiological effects in other tissues, including the cancer cells (Gaunitz and Hipkiss, 2012; Boldyrev et al., 2013; Artioli et al., 2019). However, the physiological and biochemical mechanisms responsible for the anti-neoplastic activity of carnosine are not totally known (Gaunitz and Hipkiss, 2012). It has been suggested that the primary mechanism for the anti-neoplastic effects of carnosine relates to its ability to inhibit glycolysis by its carbonyl quenching ability and therefore to reduce the generation of ATP (Boldyrev et al., 2013; Artioli et al., 2019).

In this context, we infer that the reduction in muscle carnosine may be associated with a possible targeting of this metabolite to the tumour, in order to perform its anti-neoplastic activity. Corroborating this exploration, we observed an approximate 25% reduction in the value of muscle carnosine in W21 group (mean ± SD: 2.5533 ± 1.5661) in comparison to C21 group (mean ± SD: 3.3680 ± 1.9179), although no statistically significant difference, and about 98% reduction in WL21 group (mean ± SD: 0.0514 ± 0.0248) in comparison to C21 group (mean ± SD: 3.3680 ± 1.9179), with statistically significant difference. Despite this profound significance, showing that leucine potentiated this reduction, it could not be sustained alone at this point, because there is no difference in the W21 tumour mass (mean ± SD: 35.04 ± 3.49) in comparison to WL21 group (mean ± SD: 38.8 ± 2.23). Most carnosine that reach the bloodstream, where hydrolytic enzyme is highly present and active, is immediately hydrolysed into beta-alanine and histidine (Perim et al., 2019). For this reason, we believe that carnosine was unable to reach the tumour and perform its antiproliferative role.

In conclusion, we would like to emphasize that in fact carnosine reduction is not beneficial to muscle. However, we emphasize that leucine, methionine and glucose-6-phosphate metabolites were increased in the WL21 group and that they represent benefits to the muscle. We believe that further studies on muscle and tumour are needed to better understand the role of carnosine.

Referee comment: The authors did not describe proliferation assay and MTT assay (Fig 3a) in Method.

Answer: We have included the methodology in Methods section, item 4.6, highlighted in yellow.

References

Artioli, G.G.; Sale, C.; Jones, R.L. Carnosine in health and disease. Eur J Sport Sci 2019, 19, 30-39.

Boldyrev, A.A.; Aldini G.; Derave, W. Physiology and pathophysiology of carnosine. Physiol Rev 2013, 93, 1803-1845.

Choudry, H.A.; Pan, M.; Karinch, A.M.; Souba, W.W. Branched-chain amino acid-enriched nutritional support in surgical and cancer patients. J. Nutr. 2006, 136, 314S–8S.

Cruz, B.; Oliveira, A.; Gomes-Marcondes, M.C.C. L-leucine dietary supplementation modulates muscle protein degradation and increases pro-inflammatory cytokines in tumour-bearing rats. Cytokine 2017, 96, 253–260, doi:10.1016/j.cyto.2017.04.019.

Garlick, P.J. The role of leucine in the regulation of protein metabolism. J. Nutr. 2005, 135, 1553S–6S.

Gaunitz, F.; Hipkiss, A.R. Carnosine and cancer: a perspective. Amino Acids 2012, 43, 135-142.

Gonçalves, E.M. Gomes-Marcondes, M.C.C. Leucine affects the fibroblastic Vero cells stimulating the cell proliferation and modulating the proteolysis process. Amino Acids 2010, 38, 145–153, doi:10.1007/s00726-008-0222-7.

Hardee, J.P.; Montalvo, R.N.; Carson, J.A. Linking Cancer Cachexia-Induced Anabolic Resistance to Skeletal Muscle Oxidative Metabolism. Oxid. Med. Cell. Longev. 2017, 2017, 1–14, doi:10.1155/2017/8018197.

Kimball, S.R.; Jefferson, L.S. Regulation of protein synthesis by branched-chain amino acids. Curr. Opin. Clin. Nutr. Metab. Care 2001, 4, 39–43.

Kliewer, K.L.; Ke, J.-Y.; Tian, M.; Cole, R.M.; Andridge, R.R.; Belury, M.A. Adipose tissue lipolysis and energy metabolism in early cancer cachexia in mice. Cancer Biol. Ther. 2015, 16, 886, doi:10.4161/15384047.2014.987075.

Matthews, J.J.; Artioli, G.G.; Turner, M.D.; Sale, C. The physiological roles of carnosine and β-alanine in exercising human skeletal muscle. Med Sci Sports Exerc 2019, 51, 2098-2108.

Perim, P.; Marticorena, F.M.; Ribeiro, F.; Barreto, G.; Gobbi, N.; Kerksick, C.; Dolan, E.; Saunders B. Can the skeletal muscle carnosine response to beta-alanine supplementation be optimized? Front Nutr 2019, 6.

Peters, S.J.; Van Helvoort, A.; Kegler, D.; Argilès, J.M.; Luiking, Y.C.; Laviano, A.; Van Bergenhenegouwen, J.; Deutz, N.E.P.; Haagsman, H.P.; Gorselink, M.; et al. Dose-dependent effects of leucine supplementation on preservation of muscle mass in cancer cachectic mice. Oncol. Rep. 2011, 26, 247–254, doi:10.3892/or.2011.1269.

Vaitkus, J.A.; Celi, F.S. The role of adipose tissue in cancer-associated cachexia. Exp. Biol. Med. 2017, 242, 473, doi:10.1177/1535370216683282.

Viana, L.R.; Canevarolo, R.; Luiz, A.C.P.; Soares, R.F.; Lubaczeuski, C.; Zeri, A.C. de M.; Gomes-Marcondes, M.C.C. Leucine-rich diet alters the (1)H-NMR based metabolomic profile without changing the Walker-256 tumour mass in rats. BMC Cancer 2016, 16, 764, doi:10.1186/s12885-016-2811-2.

Viana, L.R.; Gomes-Marcondes, M.C.C. A leucine-rich diet modulates the tumor-induced down-regulation of the MAPK/ERK and PI3K/Akt/mTOR signaling pathways and maintains the expression of the ubiquitin-proteasome pathway in the placental tissue of NMRI mice. Biol. Reprod. 2015, 92, doi:10.1095/biolreprod.114.123307.

Reviewer 3 Report

Very interesting paper. Relevant information is provided by authors.

It seems mandatory to add statistical comparisons of tumor evolution in the different groups to affirm/infirm the hypothesis that diet supplementation could also affect tumor growth. According to the data presented by authors, it would seem that leucine supplementation affects positively tumor growth. If this is the case this should be underlined and discussed. If this does not lower the scientific quality of the research nor the leading message, information on effect of diet on tumor is at least as important as effect on muscle.

Author Response

Referee#3

Very interesting paper. Relevant information is provided by authors.

We would like to thank Reviewer #3 for the time spent in the revision of our manuscript and we do appreciated your comment. We are certain that the revised manuscript including your suggestions gained more strengh.  The changes made in the revised paper are highlighted in yellow.

Referee comment: It seems mandatory to add statistical comparisons of tumor evolution in the different groups to affirm/infirm the hypothesis that diet supplementation could also affect tumor growth. According to the data presented by authors, it would seem that leucine supplementation affects positively tumor growth. If this is the case this should be underlined and discussed. If this does not lower the scientific quality of the research nor the leading message, information on effect of diet on tumor is at least as important as effect on muscle”

Answer: We agree that the effects of the leucine supplementation in the tumour are as important as the effects on muscle. As present in the previous version of the manuscript (Table 1. Morphometric parameters), we did not find any statistical differences in tumour weight among the experimental groups. Considering the great importance in elucidating the effects of leucine supplementation in tumour growth, we have recently published a manuscript (Viana, Tobar et al. 2019) evaluating the effects of the administration of a leucine-rich diet on the Walker 256 tumour metabolism. As presented here, in our previous study the tumour weight was similar between tumour-bearing rats and tumour-bearing rats fed with leucine rich-diet. In spite of the similar tumour weights here and in others, in our previous study we found that the leucine diet led to a tumour metabolic shifting from glycolytic towards oxidative phosphorylation, reducing glucose consumption and metastasis formation, suggesting that the leucine nutritional supplementation does not benefit this type of tumour. These points are now included in the discussion section, and highlighted in yellow.

Reference:

Viana, L. R., N. Tobar, E. N. B. Busanello, A. C. Marques, A. G. de Oliveira, T. I. Lima, G. Machado, B. G. Castelucci, C. D. Ramos, S. Q. Brunetto, L. R. Silveira, A. E. Vercesi, S. R. Consonni and M. C. C. Gomes-Marcondes (2019). "Leucine-rich diet induces a shift in tumour metabolism from glycolytic towards oxidative phosphorylation, reducing glucose consumption and metastasis in Walker-256 tumour-bearing rats." Sci Rep 9(1): 15529.

Round 2

Reviewer 2 Report

Unfortunately, my suggestions have not addressed properly.

The muscle proteomic work may be worth publishing but overall significance is equivocal.

Because it is surprising to me that increasing leucine from 1.6% to 4.6% had muscle preserving effect in their model, I asked there is any dose response relationships between this effect and leucine content. If they test no leucine(0%), plus 1.6%, 3%, 4.6% leucine diet, it would be much more interesting. The authors only mentioned their “essays” and did not tell when they started to observe this effect or any details. Potential readers will not even know the presence of their essays since it is not mentioned at all in the revised manuscript.

They should describe in the manuscript (not just in referee’s response) that all groups took their chows (per body weight) equally.

Comparing C vs L and W vs WL, it seems to me that some effects of leucine were only seen in the context of Walker256 tumor. Thus, I raised the point that how leucine’s effects seen in this model are applicable to cancer in general. Again, they referred to their “essay” on MAC16, on which potential readers or I have no idea. Although Walker256 tumor did not increase proliferation in response to leucine supplementation, some cancers do proliferate possibly due to mTOR stimulation by leucine. If leucine is beneficial even in case of “leucine-dependent tumors”, the significance would be totally different. This generalizability limitation should be described in the manuscript.

According to authors’ previous reports, leucine should modulate inflammatory cytokines and/or proteasome pathways. It is very odd to me that proteomic analysis did not pick up any inflammation or ubiquitin-proteasome related pathways in Table 3. Did they specifically search these proteins in their analysis?

Author Response

Referee #2

Answers about comments and suggestions for authors

Once again, we would like to thank the Referee for the constructive criticism. We believe the inclusions made after these comments improved the work being more assertive in its aim. We hope this new and improved version of the manuscript is now suitable for publication

In the first comment made by Referee we find the following:

Referee comment: The muscle proteomic work may be worth publishing but overall significance is equivocal.

Answer: We would like to clarify that the general objective of our manuscript is to show that energetic pathways, mitochondria, signalling proteins, mTOR, among others, are possible targets in this study using nutritional supplementation and cachexia. We intended to direct this work to contribute to the construction of knowledge in this area. Therefore, we do believe that the work brings relevant results for researchers who want to explore new approaches in the cancer-cachexia system and seek results that have some basis to follow a path with future perspectives.

Referee comment: Because it is surprising to me that increasing leucine from 1.6% to 4.6% had muscle preserving effect in their model, I asked there is any dose response relationships between this effect and leucine content. If they test no leucine (0%), plus 1.6%, 3%, 4.6% leucine diet, it would be much more interesting. The authors only mentioned their “essays” and did not tell when they started to observe this effect or any details. Potential readers will not even know the presence of their essays since it is not mentioned at all in the revised manuscript.

Answer: We have added some explanation about the use of 3% of leucine in the diet in our experiments in the Material and Methods section in the revised text of the manuscript, lines 466-470. Additionally, we have performed dose-response tests with 1.5, 3.0, 4.5% and 6% leucine added to the control diet that has 1.6% leucine, and we have gotten a positive effect with 3% leucine supplementation show in all morphometric and biochemical parameters including skeletal muscle. Above 4.5%, we did not observe any additional increase in positive effects, on the contrary, there were negative effects in renal function of animals subjected to excess leucine. These data are not published. We also have to ponder that the use of Wistar rats with the Walker 256 Tumour is a very acute cachexia model and the animals can die in about 25 days after tumour implantation, then when there is a 3% Leucine supplementation we are getting a significant interference on the signalling and modulation of the mTOR. Thus, our work aimed to show possible paths and not to conclude about effective treatments against cachexia. Therefore, through our previous results, we were able to show impacted pathways, modulation in proteins of interest, important signalling when increasing leucine in 3%.

Referee comment: They should describe in the manuscript (not just in Referee’s response) that all groups took their chows (per body weight) equally.

Answer: We fully agree with these comments, and this description has been included in the revised text of the manuscript, lines 471-476

Referee comment: Comparing C vs L and W vs WL, it seems to me that some effects of leucine were only seen in the context of Walker256 tumor. Thus, I raised the point that how leucine’s effects seen in this model are applicable to cancer in general. Again, they referred to their “essay” on MAC16, on which potential readers or I have no idea. Although Walker256 tumor did not increase proliferation in response to leucine supplementation, some cancers do proliferate possibly due to mTOR stimulation by leucine. If leucine is beneficial even in case of “leucine-dependent tumors”, the significance would be totally different. This generalizability limitation should be described in the manuscript.

Answer: We thanked the referee comments raising this important issue. We have added some paragraph referring to this point that leucine-induced mTOR stimulation in muscle tissue but also leads to important effects in this type of tumour – Walker 256. Also, the main intention of our research was to show some improvements in the host carcass, specifically in skeletal muscle (even trying this experimental model to be translated to patients). We have added a statement referring to the limitation of experimental work. As present in the previous version of the manuscript (Table 1. Morphometric parameters), we did not find any statistical differences in tumour weight among the experimental groups. Despite not increasing the tumour mass, leucine supplementation can act by stimulating the mTOR pathway, which may be a limitation to be applied to other types of tumour, even in patients. Our previous work, we showed that leucine stimulated the mTOR expression in Walker tumour tissue, without any additional growth (Viana et al, 2018). However, as reported by Nakano and colleagues leucine and other branched-chain amino acids increased cell senescence induced by DNA damage mediated by the mTORC1 pathway to regulate p21 translation (Nakano et al, 2013), which could potentially explain the lack of increase proliferation in response to leucine supplementation. Also, considering the great importance in elucidating the effects of leucine supplementation in tumour growth, we have recently published a manuscript (Viana, Tobar et al. 2019) evaluating the effects of the administration of a leucine-rich diet on the Walker 256 tumour metabolism. As presented here, in our previous study, the tumour weight was similar between tumour-bearing rats and tumour-bearing rats fed with leucine rich-diet. Besides, despite having similar tumour weights, as presented here, in our previous study we found that the leucine diet led to a tumour metabolic shifting from glycolytic towards oxidative phosphorylation, reducing glucose consumption and metastasis formation, suggesting that the leucine nutritional supplementation does not benefit this type of tumour. As our knowledge, there are no clinical studies focused on leucine supplementation in cancer tissue, in patients. These points are now included in the discussion section and highlighted in yellow in lines 289-303.

Reference:

Viana, L. R., N. Tobar, E. N. B. Busanello, A. C. Marques, A. G. de Oliveira, T. I. Lima, G. Machado, B. G. Castelucci, C. D. Ramos, S. Q. Brunetto, L. R. Silveira, A. E. Vercesi, S. R. Consonni and M. C. C. Gomes-Marcondes (2019). "Leucine-rich diet induces a shift in tumour metabolism from glycolytic towards oxidative phosphorylation, reducing glucose consumption and metastasis in Walker-256 tumour-bearing rats." Sci Rep 9(1): 15529.

Viana LR, Canevarolo R, Luiz AC, Soares RF, Lubaczeuski C, Zeri AC, Gomes-Marcondes MC.Leucine-rich diet alters the 1H-NMR based metabolomic profile without changing the Walker-256 tumour mass in rats. BMC Cancer. 2016 Oct 3;16(1):764

Nakano M, Nakashima A, Nagano T, Ishikawa S, Kikkawa U & Kamada S 2013 Branched-chain amino acids enhance premature senescence through mammalian target of rapamycin complex I-mediated upregulation of p21 protein. PLoS One 8 e80411.

Referee comment: According to authors’ previous reports, leucine should modulate inflammatory cytokines and/or proteasome pathways. It is very odd to me that proteomic analysis did not pick up any inflammation or ubiquitin-proteasome related pathways in Table 3. Did they specifically search these proteins in their analysis?

Answer: Once again, we thanked the Referee for this observation. Therefore, the proteomic analysis was directed to pathways involving mitochondrial and energetic processes, focusing the intention of advancing the questions about energy expenditure in the skeletal muscle, seeking new metabolic pathways and proteins involved in these pathways. Unfortunately, we did not direct our search to look for cytokines, cytokine receptors in muscle or proteins of the ubiquitin-proteasome system.

Reviewer 3 Report

Authors answered the questions of my reviewing.

Author Response

Once again, we thank the Referee for the time spent in very constructive analyses of our manuscript, as well as for the suggestions received, which certainly improved this work.We are glad to know that the reviewer has considered our responses as satisfactory.

Round 3

Reviewer 2 Report

This time, the authors had responded to my concerns. Only issue is the generalizability. However in the authors' response they wrote, "We also have to ponder that the use of Wistar rats with the Walker 256 Tumour is a very acute cachexia model and the animals can die in about 25 days after tumour implantation, then when there is a 3% Leucine supplementation we are getting a significant interference on the signalling and modulation of the mTOR. '' Now I understand the specific characteristics of this Walker 256 tumor. It would be good if they can present the effect of leucine on another tumor bearing model, but if not possible please include above-mentioned characteristics of Walker tumor model somewhere in he text so that readers know the justification of using this model in the study.

Author Response

Referee #2

Answers about comments and suggestions for authors

Once again, we would like to thank the Referee for all the time spent analysing our work and for all constructive criticism. We hope this new version of the manuscript is now suitable for publication

This time, the authors had responded to my concerns. Only issue is the generalizability. However in the authors’ response they wrote, “We also have to ponder that the use of Wistar rats with the Walker 256 Tumour is a very acute cachexia model and the animals can die in about 25 days after tumour implantation, then when there is a 3% Leucine supplementation we are getting a significant interference on the signalling and modulation of the mTOR. “ Now I understand the specific characteristics of this Walker 256 tumor. It would be good if they can present the effect of leucine on another tumor bearing model, but if not possible please include above-mentioned characteristics of Walker tumor model somewhere in he text so that readers know the justification of using this model in the study.

Answer: We have now added in Introduction section (lines 79-83, highlighted in yellow) the statement explaining a little more about the characteristics of this experimental model of cachexia, where we do believe it is worth working on.

“Thus, it is known that the Walker-256 tumour has an exponential growth, which can reduce the survival time, killing the bearing rat approximately 25 days after the tumour implantation; then, when there was a 3% leucine supplementation, a significant interference could be observed in cell signalling and modulation of mTOR pathway in the host, especially in the gastrocnemius muscle [14].”